# Feeding Forage Mixtures of Ryegrass (*Lolium* spp.) with Clover (*Trifolium* spp.) Supplemented with Local Feed Diets to Reduce Enteric Methane Emission Efficiency in Small-Scale Dairy Systems: A Simulated Study

**DOI:** 10.3390/ani11040946

**Published:** 2021-03-27

**Authors:** Maria Danaee Celis-Alvarez, Felipe López-González, Carlos Manuel Arriaga-Jordán, Lizbeth E. Robles-Jiménez, Manuel González-Ronquillo

**Affiliations:** 1Instituto de Ciencias Agropecuarias y Rurales, Universidad Autónoma del Estado de México, No. 100 Instituto Literario, Toluca 50000, Estado de México, Mexico; mvzdanaeecelis@yahoo.com.mx (M.D.C.-A.); cmarriagaj@uaemex.mx (C.M.A.-J.); 2Facultad de Medicina Veterinaria y Zootecnia, Universidad Autónoma del Estado de México, No. 100 Instituto Literario 100, Col. Centro, Toluca 50000, Estado de México, Mexico; lizroblez@hotmail.com

**Keywords:** enteric methane emissions, supplementation, feeding strategies, dairy

## Abstract

**Simple Summary:**

The present study simulated the effects of different dairy cow diets based on local feeding strategies on enteric methane (CH_4_) emissions and surpluses of crude protein (CP) in small-scale dairy systems (SSDS). Our study evaluated five scenarios of supplementation (S): without supplementation (control diet)*,* meaning no supplements were provided, only pasture (S1); pasture supplemented with 4.5 kg dry matter (DM)/cow/day of commercial concentrate (CC) (S2); supplemented with 200 g DM/kg per milk produced of CC (S3); supplemented with ground maize grains and wet distiller brewery grains (S4); and S4 plus maize silage (S5). In addition, two pasture managements (cut-and-carry versus grazing) and two varieties of legumes (red clover vs. white clover) were considered. The results suggest that methane emissions and large nitrogen surpluses in the diet are affected by the type of supplementation given to cows, in addition to the management and chemical composition of the pastures offered. In SSDS, it is possible to formulate diets with local inputs to reduce excess nutrients and dependence on external inputs, increasing feed efficiency and reducing costs (excess of CP in the diet) and CH_4_ emissions.

**Abstract:**

In cattle, greenhouse gas (GHG) emissions and nutrient balance are influenced by factors such as diet composition, intake, and digestibility. This study evaluated CH_4_ emissions and surpluses of crude protein, using five simulated scenarios of supplementation in small-scale dairy systems (SSDS). In addition, two pasture managements (cut-and-carry versus grazing) and two varieties of legumes (red clover vs. white clover) were considered. The diets were tested considering similar milk yield and chemical composition; CH_4_ emission was estimated using Tier-2 methodology from the Intergovernmental Panel on Climate Change (IPCC), and the data were analyzed in a completely randomized 5 × 2 × 2 factorial design. Differences (*p* < 0.05) were found in predicted CH_4_ emissions per kg of milk produced (g kg^−1^ FCM 3.5%). The lowest predicted CH_4_ emissions were found for S3 and S4 as well as for pastures containing white clover. Lower dietary surpluses of CP (*p* < 0.05) were observed for the control diet (1320 g CP/d), followed by S5 (1793 g CP/d), compared with S2 (2175 g CP/d), as well as in cut-and-carry management with red clover. A significant correlation (*p* < 0.001) was observed between dry matter intake and CH_4_ emissions (g^−1^ and per kg of milk produced). It is concluded that the environmental impact of formulating diets from local inputs (S3 and S4) can be reduced by making them more efficient in terms of methane kg^−1^ of milk in SSDS.

## 1. Introduction

In 2013, greenhouse gas (GHG) emissions in Mexico were 665 Mt Carbon Dioxide Equivalent (CO_2_ eq). Emissions from the agri-livestock sector represented 12% of emissions at the national level, with the main source of methane (CH_4_) emissions being from enteric fermentation (63.9%) and, to a lesser extent, manure management (4.2%). For nitrous oxide (N_2_O) emissions, approximately one-third (31.4%) were associated with manure and agricultural soil management [1].

The diet of dairy cows is a determining factor in GHG emissions; at the same time, it is necessary to formulate diets to fulfill the nutritional requirements of dairy cows during lactation, considering the availability of different feeds throughout the year [2]. In addition, the minimization of final byproducts is also sought to reduce the environmental impact. The phosphorus content of manure is also associated with the contamination of tanks and streams, promoting eutrophication—a consequence that can be minimized through balanced rations [3] in dairy cattle.

Of the total gross energy intake (GEI) by dairy cows, 4% to 7% is lost as CH_4_ [4]. Enteric CH_4_ emissions and feces management differ based on the physiological state of cows, but overall, emissions are negatively correlated with feed utilization efficiency [5].

If diets are not adequately formulated, the overfeeding of crude protein (CP) increases feeding costs and also causes environmental problems. A large amount of nitrogen (N) may then be wasted through feces and urine. In small-scale dairy systems (SSDS), Pozo-Leyva et al. [6] found a positive N balance, indicating an excess of N in the diets of cattle. To reduce GHG emissions and N_2_ O in dairy production, Jayasundara et al. [7] mentioned that it is necessary to design appropriate feeding strategies and to implement a manure management chain. However, among small producers, there is resistance to reducing the amount of feed rich in CP in dairy cattle diets. Mostly, CP is supplied through commercial concentrates (>20% CP) at levels of 3 to 5 kg dry matter (DM)/animal/day in SSDS [8,9]. It is therefore recommended that animals graze pastures associated with legumes in order to reduce feed costs and cover their nutritional requirements; however, they often do not cover their energy or protein requirements, and hence farmers look for supplementation strategies with commercial concentrates or local byproducts, which reduce costs and allow them to cover their nutritional requirements throughout the year.

Given this context, the hypothesis addressed in the present study was that it is feasible to reduce the emission of CH_4_ per kilogram of milk and reduce the excretion of nitrogen and phosphorus, and thus reducing the environmental footprint of these systems.

In this study, we aimed to evaluate, through simulation, the effects of different dairy cow diets grazing two different pastures management and two varieties of legumes (red clover vs. white clover) with the inclusion as supplements of common local ingredients used in SSDS on the emission intensity of CH_4_ per kilogram of milk and on nitrogen and phosphorus intake and excretion.

## 2. Materials and Methods

### 2.1. Study Area

The simulation study was based on previous experience in the municipality of Aculco, State of México, México (between 20° 05′57″ N and 99° 49′41″ W, 2428 m above sea level) with a sub-humid temperature climate. The mean temperature during the years 2011–2019 was 16 °C, with a mean minimum temperature of 7 °C and a mean maximum temperature of 24 °C, and 888 mm of rainfall (Figure 1).

Tmax: Maximal Temperature (°C), Tmin: Minimal temperature (°C), and rainfall (mm) in the study area. The data were obtained from the meteorological station “La Concepcion” (No. 15189) by CONAGUA-DGe.

Dairy production is one of the main activities in Aculco, State of Mexico, where 82% of milk is destined for artisanal cheese production. The study area is characterized by SSDS, with a typical herd size of between 3–20 animals [9,10]. Martínez-García et al. [11] in their study considering this criterion (data were collected from 115 farmers; the sample size represents 5% of the total farms in the study area), found an average herd size of 10 cows, of which only 4.4 cows were in production. In other countries, the herd size of SSDS is 1–10 cows in intensive and extensive systems (East Central Africa) [12] or 20 animals in all reproductive stages (Central Colombia) [13]. Fadul-Pacheco et al. [14] previously characterized the SSDS in the study area, and Alfonso-Ávila et al. [8], Martínez-García et al. [9], Celis-Alvarez el al. [15], López-González et al. [16], Jaimez-García et al. [17], Sainz-Sánchez et al. [18], Plata-Reyes et al. [19], Becerril-Gil et al. [20]; Burbano-Muñoz et al. [21], Carrillo-Hernández et al. [22], Gómez-Miranda et al. [23], González-Alcántara et al. [24], López-González et al. [25], Marín-Santana et al. [26], Rosas-Dávila et al. [27], Vega-García et al. [28], Muciño-Álvarez et al. [29], identified different local feeding strategies used during the rainy and dry seasons. Based on previous research studies, the different feeding scenarios were determined (Table 1).

### 2.2. Diet Specifications

The present simulation with dairy cows and research was performed previously with collaborating farmers in previous studies [8,9,17,18,25], following the procedures of the Universidad Autónoma del Estado de México. To determine the diet specifications, the following variables were kept constant: live weight (LW), days of lactation, milk yield, and chemical composition (Table 2) in all treatments. The data were obtained from previous research studies in the region [8,9,30,31]. Briefly, samples of pastures (red clover (RC) (*Trifolium pratense)* with perennial ryegrass (*Lolium perenne*) and white clover (WC) (*Trifolium repens*) with annual ryegrass (*Lolium multiflorum*), commercial concentrate, ground maize grains (GMG), and wet distillers’ grains (WDG), were dried in a forced-air oven at 60 °C for 48 h. Once dried, they were ground with a Wiley mill (2.0 mm screen; Arthur H. Thomas, Philadelphia, PA, USA) and analyzed in duplicates for DM (930.15) and nitrogen (N; 990.02) using the Association of Official Analytical Chemists (AOAC) [32] standard methods. Neutral detergent fiber (NDF) and acid detergent fiber (ADF) were determined following Van Soest et al. [33] methods. In vitro dry matter digestibility (IVDMD) was performed of each ingredient by the Ankom–Daiy method [17,34]. Other details on diet preparation, including the control and supplementation strategies, were described previously [8,9,30,31].

### 2.3. Diet Formulation

The chemical composition of feeds was obtained from information collected for previous research studies in the region during the dry season [9,30] and rainy season [8,31]; missing data were obtained using data from the National Research Council (NRC) [35] and the Institut National de la Recherche Agronomique (INRA) [36]. The diets were formulated using two pasture management strategies (M): grazing of cultivated pastures (G), which has been a successful technology adopted by farmers to reduce feeding costs [30], and cut-and-carry (C), a conventional feeding strategy used by the SSDS [31]. The two pasture management strategies (G and C) were composed of different clover varieties (Var): RC (*Trifolium pratense*) with perennial ryegrass (*Lolium perenne*) and WC (*Trifolium repens*) with annual ryegrass (*Lolium multiflorum*). The four combined diets were then formulated with different supplementation strategies and evaluated using the followed combinations: G/RC, G/WC, C/RC, and C/WC, using 15 cows per diet (Table 1). The criterion used, standardizing the number of cows, the breed, and their milk production, was used so that there would be no significant differences between animals and milk production, and the results would focus on feeding strategies based on previous studies (Table 1 and Table 2) and other suggested as the inclusion of wet distillers grains.

For each diet, the quantities of Metabolizable Energy (ME, MJ/d), Crude Protein (CP, g/d), rumen degradable protein (RDP,g/d), neutral detergent fiber (NDF,g/d), Ca (g/d), and P (g/d) required for maintenance, gestation, and lactation were estimated.

### 2.4. Animals and Diets: Scenarios of Supplementation

After the chemical composition of the ingredients and nutritional requirements of the cows of each diet was calculated, we simulated performing five diet scenarios of supplementation (S).

Supplementation 1: No supplements were provided. Total dry matter intake (TDMI, Kg/d) only consisted of that obtained from the two pastures managements (G and C); Supplementation 2: The two pastures managements (G and C) supplemented with 4.5 kg DM per cow per day of a commercial concentrate (19% CP) commonly used as a diet component in local feeding strategies was provided [11]; Supplementation 3: The two pasture management strategies (G and C) supplemented with 200 g DM per kg of milk produced by cows using a commercial concentrate (19% CP), commonly used by SSDS in the region, was provided; Supplementation 4: nutritional requirements were covered using the two pasture management strategies (G and C) supplemented with ground maize grains (GMG) and wet distillers’ grains (WDG). The prior is rich in non-fibrous carbohydrates and the latter in CP. Both are commonly utilized in the region [8]; and Supplementation 5: The two pasture management strategies (G and C) supplemented with maize silage (MS), commonly used during the dry season [9], was added to grasses in a 50:50 ratio in addition to supplementation with GMG and WDG to cover the nutritional requirements for the dairy cows in the present study.

### 2.5. Calculations

The metabolizable energy (ME) requirements for maintenance, pregnancy, milk yield, and the live weight change of cows was calculated with the formulas proposed by the Ministry of Agriculture, Fisheries, and Food [37], using the following equations:MEm (MJ/d) = 8.3 + 0.091 × LW(1)
where LW = live weight (kg)
MEp (MJ/d) = 1.13 e ^0.0106 t^,(2)
where t = number of days pregnant.

The prediction of total DM intake (TDMI) and Fat corrected milk (FCM) was calculated [35] using the following equation
TDMI (kg/d) = 0.372 × FCM 0.0968 × LW^0.75^) × (1 − e ^0.192^ × ^(WOL +3.67))^),(3)
where FCM = 3.5 percent fat-corrected milk (k/d), and WOL = week of lactation
3.5% FCM (kg/d) = [0.432 × milk yield (kg)] + [16.23 × milk fat yield (kg)],(4)

The calcium (Ca) and phosphorus (P) required for maintenance (m), gestation (g), and lactation (l) of dairy cows were calculated as follows [35]
Ca_m_ (g/d) = 0.031 g/kg × LW(5)
where LW = live weight (kg)
Ca_g_ (g/d) = 0.02456 e((0.05581 − 0.00007 (t))(t)) − 0.02456 e((0.05581 − 0.00007(t − 1))(t − 1)),(6)
where t = day of gestation
Ca_l_ (g/kg milk) = 1.37 g for other breeds,(7)
P_m_ = 1 g/kg DM × total DM fed + 0.002 × LW,(8)
where DM = dry matter and LW = live weight (kg)
P_g_(g/d) = 0.02743 e ((0.05527 − 0.000075 (t))(t)) − 0.02743 e((0.05527 − 0.000075 (t − 1))(t − 1))(9)
where t = day of gestation:P_l_ (g/kg milk) = 0.9 g × milk yield(10)

Forage intake was restricted using the Forage Unit for milk production [36], dividing the amount of DM intake (DMI) by kg LW^0.75^ of the foraging pattern for the considered feed. The P and Ca requirements per productive stage were also determined.

### 2.6. Estimation of Enteric Methane Emissions

To estimate CH_4_ emission from enteric fermentation, tier 2 equations proposed by the Intergovernmental Panel on Climate Change (IPCC) [38] were used considering a conversion factor (Y_m_) of 6.5% for dairy cattle.

Methane (CH_4_) production was calculated from the GEI (MJ head^−1^ day^−1^) of concentrate and grass intake. Daily methane production was calculated based on the IPCC [38]
CH_4_ (g/d) = (GEI × (Y_m/_100))/55.65(11)
where CH_4_ = methane emission (g-head^−1^-day^−1^); Y_m_ is the percentage of GE converted to methane calculated
Y_m_ = (6.5% of GEI)(12)
where GEI = gross energy intake (MJ-head^−1^-day^−1^)
GEI = (NE req/REM)/(DE/100)(13)
where NE req = summed net energy requirements (maintenance, lactation, and pregnancy), REM = ratio of net energy available in a diet for maintenance to digestible energy consumed, and DE = gross energy (MJ day^−1^).

### 2.7. Statistical Analysis

A completely randomized 5 × 2 × 2 factorial design was used, including the different supplementations scenarios (n = 5), two pasture managements (n = 2), and two pasture composition (different varieties of legumes WC/RC; n = 2). Sixty replicates (cows) were used out for the evaluation of the scenarios along with the following interactions: scenario × management, scenario × variety, management × variety, and scenario × management × variety. The significant differences (*p* < 0.05) were evaluated by Tukey mean comparisons.

Subsequently, a Pearson correlation analysis was carried out to estimate the relationship between the variables of LW^0.75^, DMI, NDF intake, N intake, ME intake, Milk Yield, FCM, CH_4_, and to assess the association among nutrient intake, CH_4_ of different strategies, five different supplementations (S), two pasture systems (grazing or cut-and-carry), and two forage crops (red clover or white clover), using the statistical program Minitab V17 (Minitab LLC, State College, PA, USA) [39].

## 3. Results

### 3.1. Chemical Composition of Ingredients

The chemical composition of the ingredients considered in the diets is shown in Table 3. The cut pastures (C/RC, C/WC) had the lowest amount of CP, highest fiber content, and lowest digestibility compared with grazing pastures. Therefore, grazing pastures had better nutritional quality. The GMG and WDG had higher non-fibrous carbohydrate content and protein content, respectively, with respect to the forages, in addition to better digestibility. The WDG also had the highest P content and ME (MJ/kg DM) compared to the rest of the ingredients.

### 3.2. Scenarios of Supplementation

The different scenarios of supplementation affected (*p* < 0.05) the DMI (g/kg LW^0.75^). After adjusting the supplementation level per kg/cow/day (S2), supplement intake increased and, therefore, DMI (g/kg LW ^0.75^/cow/day) (Table 4). As shown in Table 4, the balance of the nutritional requirements in regard to ME and RDP in S4 and S5 also affected (*p* < 0.05) the energy intake compared to the rest of the scenarios. S2 was associated with a greater intake of CP (*p* < 0.05) due to the distinct protein contribution of this diet, and an excess of CP was observed across all diets. However, as presented in Table 5, a decrease in excess of CP was found for S5 (*p* < 0.05).

In the estimation of CH_4_ emissions, significant differences (*p* < 0.05) were found among the simulated supplementation scenarios. The predicted CH_4_ emissions (g kg^−1^ FCM 3.5%) decreased per kg of milk produced from S3 and S4, followed by S2 and S5, which also had greater milk production than S1 (control diet) (Table 6). In addition, the lowest emissions of g CH_4_/day were similar in S3 and S4, where S4 used GMG and WDG as supplements, and where S3 used commercial concentrate based on kg of milk produced as a supplement (Table 5).

### 3.3. Pasture Management

Intake among supplements had an effect on pasture management (*p* < 0.05). DMI (g/kg LW^0.75^/cow day) was similar (*p* > 0.05). With respect to pasture management (Table 4), a greater intake of supplements was observed in C with respect to G *(p <* 0.05). Higher intake (*p* < 0.05) of Ca and NDF in addition to a lower intake of CP, RDP, and ME was observed for C compared with G systems management (Table 6).

G was associated with the lowest CH_4_ emissions (g kg^−1^ of FCM 3.5%) compared to C (*p* < 0.05).

### 3.4. Pasture Varieties

White Clover was associated with the largest intake of CP (*p* < 0.05) and an excess of nutrients (*p* < 0.05), specifically CP and RDP, with respect to red clover (Table 7). In addition, the lowest g CH_4_ day emissions corresponded to white clover as part of the pasture composition.

### 3.5. Correlation CH_4_

DM intake was positively correlated (*p* < 0.001) with MY (r = 0.938), FCM (r = 0.947), and negatively correlated (*p* < 0.001) with CH_4_ (g/kg^−1^ FCM 3.5%) (r = −0.740) (Table 8).

Acccording to the supplementation strategies (Table 8) correlated with CH_4_ emissions (g/d) there was an effect (*p* < 0.001) when it was correated with DMI (kg/d), S2 to S5 were positively correlated (*p* < 0.001) except S1 (*p* < 0.001) where a negatively correlation with DMI (r = 0.990) was observed.

Similarly, the pasture management and pasture varieties showed a positive correlation (*p* < 0.001) with DMI and CH_4_ emissions (g/d).

A negative correlation (*p* < 0.001) was observed between DMI and CH_4_ (g/kg FCM 3.5%) for the supplementation strategies and pasture management and pasture varieties (Table 9).

## 4. Discussion

### 4.1. Energy (MJ/ME) and Nutrient Intake (g/kg) 

In all feeding scenarios (S, M, and Var), high CP intake was observed. A larger supply of CP in the diet results in greater excretion of N [40]. In dairy cattle grazing native pastures, Sainz-Sánchez et al. [18] did not find differences in the productive response of low-production cows (11.9 kg/d) at two levels of CP supplementation (6 kg (high) and 4 kg (low) of concentrate per day). Following protein supplementation, Danes et al. [40] did not observe an improvement in production, nutrient digestion, or microbial protein synthesis in grazing dairy cattle but did observe an effect on the concentrations of plasma urea nitrogen (PUN), nitrogen urea in milk (MUN), and N-NH_3_. These authors [40] highlighted that the protein content of grass in addition to an energy supplement was sufficient for fulfilling the nutritional requirements of dairy cows. Similar results were observed in medium-production cows (20 L/d^−1^) grazing on tropical gramineous plants containing 18.5% CP [40].

Marín-Santana et al. [26], in their study, found similar milk yields (19.16 kg milk yields) as the present study, grazing ryegrass pastures and Kikuyu grass (*Pennisetum clandestinum)* associated with white clover. In another study [21], carried out in dairy cows feed with Ryegrass grazing *Festulolium* associated with white clover, oat silage, and commercial concentrate, obtained similar milk yields (18.86 kg/d/cow) as in the present study. It is important to consider that under the conditions of the present simulation study, the predictions made with previous studies coincide [15,20,21,22,23,26], considering that under SSDS systems, they are in the range of 14 to 20 kg milk production.

Specifically, the S5 diet, including maize silage, had the lowest excess CP, similar to the results of Wilkinson and Garnsworthy [2], who simulated and formulated different diets based on pasture grazing, grass, and maize silage, and their co-products. Specifically, the diets with grass and maize silage were associated with the lowest N levels and highest nitrogen use efficiency (NUE). In this sense, the diets based on maize silage had a better balance between the supply and requirement of effective rumen degradable protein (ERDP). In the present study, S5 corresponded with a 50:50 ratio of maize silage to grass. Notably, maize silage is characterized by its low concentration and digestibility of N in the rumen compared to N from grass [2]. Therefore, in the cut-and-carry strategy, there was a greater excess of CP. In dairy systems with moderate levels of milk production based on pasture feeding, such as New Zealand, higher N intakes in the diet are common due to the high content of CP in the pasture [41]. Grasses in cut-and-carry management had lower CP content compared to grazing management. Wilkinson and Garnsworthy [2] found that grass under grazing pastures presented a higher N content than cut grass due to phenological characteristics and the presence of leaves. Mainly, grass under grazing is in earlier phenological stages compared to cut grass at a later stage of maturity. As a result, a higher concentration of N can be excreted to the environment, and the NUE of animals is reduced. Therefore, the use of environmentally friendly diets allows animals to graze while reducing N emissions to the environment [42].

Low-cost supplement options are commonly used in diets. Energy is the most limiting and costly nutrient when milk production levels are low, as shown in this study where scenarios S1–S3 showed a limitation with respect to ME (MJ/d). To cover this requirement at a low cost, protein-rich supplements are often provided, yet these can result in excess protein levels in the diet [2]. Danes et al. [40] mentioned that dairy producers in Brazil face this problem when providing protein-rich supplements. Although this strategy is frequently used, excess supplementation can destabilize the balance between economic costs and environmental consequences, resulting in the elimination of large quantities of nitrogen in urine and feces [42], as we observed in the S2 strategy. Therefore, in moderate dairy production systems, the use of complementary high-protein ingredients should be reduced, and the proportion of forage in the diet increased to balance the energy to protein relationship and fulfill animals’ energy requirements.

The estimation of P in the nutritional balances of the feeding scenarios indicated an excess of this mineral and a potentially greater environmental impact in animals fed with WDG (S4 and S5) and higher proportions of commercial concentrate (S2). The average P intake was 42 g/d, less than that reported by Wu et al. [43], who suggested the inclusion of 0.38% to 0.40% P (75 to 93 g/d P), which appears to be adequate for maintaining the P balance in high-production cows (>10,000 kg per lactation). In this regard, the P content in feces and urine increases with increasing P concentrations in the diet [43]. The apparent digestibility of the P in each ingredient (not calculated in this work) should also be considered, as this could possibly influence the amount of excess P. Wu et al. [43] observed that 36% of P (digestible P) was absorbed by dairy cows; so, it might be implied that for each additional gram of P, an additional 0.64 g of P would be excreted in feces, exceeding the amount required by animals. Notably, excess of P has an important environmental impact on water resources by promoting eutrophication [3].

### 4.2. Estimation of Enteric Methane Emissions

Currently, several indirect methods exist for the estimation of enteric CH_4_, including that provided by the IPCC [5,7,42]. The level 2 calculations of the IPCC were presently used with the aim of generating useful information for proposing mitigation strategies [44].

Average enteric CH_4_ emissions per year (kg CH_4_ y^−1^) were higher than reported in dairy cows in Canada (118 kg/year) [45], but in terms of g CH_4_ per day/cow, emissions were lower in dairy cows under rotational grazing in New Zealand (402 ± 52 g/day) [46], but similar results in Ireland (357.6 ± 4 g/day) [47] and also in intensive dairy farming in China (370 g/day) [48]. Regardless of management cut and carry or grazing (366 ± 2 g/day) was similar between treatments (*p* > 0.05), being lower when using white clover (359.9 g/day) compared to red clover (372.7 g/day), the latter agrees with Carrillo Hernandez et al. [23] who found that cows that grazed annual pastures emitted a higher amount of methane (266.6 g/day), compared to cows that grazed perennial pastures (242.6 g/day); hence, it is important to consider that depending on the type of forage, enteric fermentation and therefore methane emission will vary.

Boadi et al. [42] mentioned that the fermentation velocity of carbohydrates influenced the proportion of volatile fatty acids (VFA) formed and, therefore, CH_4_ production. Notably, grain supplementation in the diet was found to reduce the production of CH_4_ per evaluated product. Meanwhile, Benchaar et al. [49] encountered a decrease in energy losses from CH_4_ after providing ME in the diet. This was associated with a lower proportion of acetate:propionate, without negative effects on fiber digestibility. Dall-Orsoletta et al. [50] observed that CH_4_ production in cows grazing annual ryegrass supplemented with oats and GMG was 321 g/day, whereas with MS was 356 g/day, similar values to the present study. The use of MS and GMG as supplements is an effective strategy for reducing CH_4_ excretion yield during grazing [51]. However, Boadi et al. [42] mentioned that net GGE should be evaluated since the use of grains in diets might imply an increase in NO_2_ emissions due to the fossil fuels, fertilizers, and machinery required to produce grains. Increasing animal productivity through integrating feeding and management strategies is also a way to reduce CH_4_ production.

Among the additional factors to consider in CH_4_ production is the maturity of grass [42]. In the present study, the cut-and-carry strategy had a lower protein content, higher fiber content, and lower digestibility. On the other hand, plants under active growth have a higher cellular content, which then declines as plants mature. In this respect, there are several factors that interact to affect CH_4_ production, including the maturity of forage, the presence of secondary compounds, and other environmental factors (i.e., season) [47].

Previously it was noted that CH_4_ emissions per kg of produced milk decrease as milk production increases [2]. Jiao et al. [51] also observed that increasing the level of supplementation (2.0, 4.0, 6.0, and 8.0 kg/cow/day) in dairy cows grazing perennial ryegrass led to increased milk production per cow per day, thereby reducing CH_4_ emissions per unit of produced milk due to the increase in soluble carbohydrates. As has been shown, an increased proportion of cereal concentrates in the diet decreases enteric CH_4_ emissions, whereas the intake of non-structural carbohydrates such as starch increases hydrogen and CH_4_ production in the rumen [52]. Accordingly, Zhu et al. [48] suggested that one option to mitigate CH_4_ emissions in dairy cows is to increase milk production through optimization of the ingredients in the diet. However, the improvement in the productivity of the diets must be evaluated in terms of economic and environmental costs.

Similar to our study, Renand et al. [53] found in beef Heifers a positive correlation between DMI with CH_4_. Methane is a byproduct of anaerobic microbial fermentation of feed in the rumen, and energy used for its synthesis is considered as a loss of energy for animal production; it has been calculated that the energy loss fluctuates between 3% and 6.5% on average for cattle fed diets high in concentrates and low-quality pastures, respectively [52]. The inclusion of supplementation strategies in the diets resulted in a significant increase (*p* < 0.01) in CH_4_ production. The methane increase was 64%, 48%, 44%, and 67%, compared S1 vs. S2, S3, S4, and S5, respectively (where 100% is equivalent to the emission of CH_4_ measured without the inclusion of supplements, S1 as a control diet), and coincided with the findings of Bell et al. [54], where the average CH_4_ concentration calculated from burping peaks was positively associated with TDMI with a limited number of animals.

Lassen et al. [55], in a study with Holstein dairy cows, found a positive genetic correlation (*p* < 0.05) between milk yield and CH_4_ production, which suggests the relationship between energy intake, CH_4_ production, and milk production, which agrees with the present results, Hegarty et al. [56] reported a 40% lower daily DMI and an analogous 25% lower methane production (g/d) between high and low-residual feed intake in steers, which is the difference in daily DMI (15%) obtained between the high- and low intakes in heifers.

In the present study, S1 presented 40% lower DMI (g/kg LW^0.75^) compared with the rest of the supplementation strategies, which corresponded with lower CH_4_ production. Total daily CH_4_ production was higher in S2 to S5 than S1; this was normal because the forage proportion in the basal diets was higher than S1 (forage: concentrate ratio was 100:0, S1). It is well established that both daily CH_4_ emissions per unit of DMI increase as a result of increased NDF intake [57]. The present results coincide with Bittante and Cecchinato [58], who correlated milk yield, which is genetically determined, with estimated daily methane production, although this is mainly because the latter is obtained from the former and because increased milk yield implies larger DMI and rumen loads.

## 5. Conclusions

In small-scale dairy systems, it is possible to formulate diets from local inputs to reduce excessive nutrients and dependence on external inputs. This practice reduces the environmental impact in terms of CH_4_ emitted per kg of milk produced. Commercial concentrate supplements, while popular with local farmers, not only cost more than local supplements but may result in excessive crude protein in the diet, leading to higher excretions of nitrogen to the environment.

The hypothesis is accepted as supplements from local inputs can fulfill the nutritional requirements of dairy cattle at moderate production levels with the use of grazing pastures.

## Figures and Tables

**Figure 1 animals-11-00946-f001:**
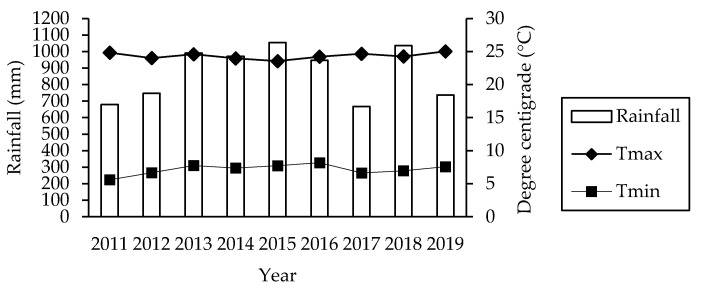
Rainfall and mean temperatures during the years 2011–2019 in the present study.

**Table 1 animals-11-00946-t001:** Research studies published previously, which were based on determining the different feeding strategy systems management and scenarios.

Author	Cows	Breed	Milk Yield (kg)	Feeding Strategy	Observations
Jaimez-García et al. [17]	10	Holstein	18.15	Maize silage, isoproteic concentrate, continuous grazing ryegrass, and white clover associated orchard, commercial concentrate.	Analysis was in a split-plot.
López-González et al. [16]	6	Holstein	15.7	Ryegrass and Festulolium grazing associated with white clover and commercial concentrate.	Latin Squares 3 × 3, randomizing cows and treatment sequences in the first square, randomizing cows in the second square.
Celis-Alvarez et al. [15]	8	Holstein	15.52	Maize silage, Ryegrass grazing associated with White clover, and commercial concentrate.	Replicated 4 × 4 Latin Square design.
López-González et al. [25]	6	Holstein	11.8	Fescue and Ryegrass grazing associated with White clover, and commercial concentrate.	Double cross-over design was applied, with three 14 day each experimental period.
Sainz-Sánchez et al. [18]	12	Holstein	11.87	Native pasture grazing and commercial concentrate.	Cows were allotted to a triple 4 × 4 Latin square design with four cows per square treatment.
Plata-Reyes et al. [19]	12	Holstein	14.85	Ryegrass, Festulolium, Fescue, and Kikuyu grazing, associated with white clover and commercial concentrate.	Cows were allotted to a triple 4 × 4 Latin square with four cows per square treatment.
Becerril-Gil et al. [20]	8	Holstein	15.90	Ryegrass cut and curry pasture associated with white clover, oat silage, and commercial concentrate.	Latin square 4 × 4, randomizing cows and treatment sequences in the first square treatment.
Burbano-Muñoz et al. [21]	9	Holstein	18.86	Ryegrass grazing, Festulolium associated with white clover, oat silage, and commercial concentrate.	Cows were allotted to a triple 3 × 3 Latin square with four cows per square treatment.
Carrillo-Hernández et al. [22]	8	Holstein	15.05	Annual Ryegrass and Perennial Ryegrass grazing associated with white clover and commercial concentrate.	Double-cross over experimental design.
Gómez-Miranda et al. [23]	9	Holstein	14.9	Ryegrass grazing associated with white clover, oat silage, and commercial concentrate.	Nine Holstein cows in groups of three were randomly assigned to a 3 × 3 Latin square design repeated three times.
González-Alcántara et al. [24]	8	Holstein	12.31	Ryegrass and Fescue grazing associated with white clover, triticale silage, and concentrate commercial.	The experimental design was 2 × 2 factorial in repeated 4 × 4 Latin squares treatment.
Marín-Santana et al. [26]	9	Holstein	19.16	Ryegrass grazing, Kikuyu associated with white clover and commercial concentrate.	Experimental 3 × 3 Latin Square design, repeated three times.
Rosas-Dávila et al. [27]	10	Holstein	13.4	Ryegrass grazing, Fescue associated with White clover and commercial concentrate.	Double crossover design with five cows per treatment and four experimental periods following the treatment sequences.
Vega-García et al. [28]	9	Holstein	15.16	Ryegrass grazing associated with White clover, black oat silage (*Avena strigosa Schreb.),* and commercial concentrate.	Latin squares 3 × 3 replicated three times.
Muciño-Álvarez et al. [29]	6	Holstein	16.75	Grazing Ryegrass, Fescue, Bromo (*Bromus willdenowii* cv. Matua), Festulolium, White clover, Red clover, and commercial concentrate.	Double cross-over experiment design.

**Table 2 animals-11-00946-t002:** Productive characteristics of cattle fed with two pasture managements (cut-and-carry pastures) and supplemented with different feeding strategies.

Cow	LW (kg/cow)	MY (kg/d) ^1^	MY (kg/d) ^2^	Milk Fat (%)	DL (d)	LW^0.75^ (kg/cow)	FCM (kg/cow/d) ^1^	FCM (kg/cow/d) ^2^
1	598	5.9	21.0	3.6	186	120.9	6.0	21.3
2	536	3.1	12.3	4.0	186	111.4	3.4	13.3
3	548	4.8	18.0	3.7	198	113.3	5.0	18.6
4	594	5.3	19.4	3.3	158	120.3	5.1	18.7
5	515	5.5	17.4	3.6	114	108.1	5.6	17.7
6	578	6.5	24.1	3.1	177	117.9	6.0	22.4
7	544	6.4	23.6	2.8	140	112.6	5.6	20.8
8	462	4.0	15	3.4	174	99.7	3.9	14.8
9	471	5.2	16.7	3.4	116	101.1	5.1	16.5
10	463	3.1	12.8	3.0	154	99.8	2.8	11.7
11	489	4.4	16.0	3.7	184	104.0	4.5	16.4
12	473	3.7	14.3	3.1	153	101.4	3.4	13.3
13	379	3.2	10.9	3.6	125	85.9	3.3	11.0
14	559	5.7	18.3	3.9	153	114.9	6.0	19.4
15	468	3.7	13.2	3.7	154	100.6	3.8	13.5
Average	512	4.7	14.0	3.4	158	107.5	4.6	16.6

LW: live weight, MY: Milk yield, DL: Days of lactation, FCM: Fat-corrected milk, FCM 3.5% (kg/d) = (0.432 × milk yield (kg)) + (16.23 × milk fat yield (kg)), ^1^ Milk yield and fat corrected milk used in scenario 1, ^2^ Milk yield and fat corrected milk used in scenarios 2, 3, 4, and 5.

**Table 3 animals-11-00946-t003:** Chemical composition (g/kg DM) of the ingredients used to shape the different feeding scenarios in dairy cows.

Item ^†^	G/RC	G/WC	C/RC	C/WC	GMG	CC	WDG	MS
DM	182	157	188	175	872	902	250	310
CP	177	193	139	145	80	190	280	78
RDP	106	116	83	87	51	114	222	47
NDF	439	403	481	489	90	273	180	477
ADF	236	218	275	283	21	93	50	275
IVDMD	705	720	675	691	863	817	850	607
Ca	7.8	7.4	7.6	8.4	0.2	13.1	3.0	2.8
P	3.9	3.6	3,4	3.7	3.0	3.7	7.8	1.6
ME, MJ/kg DM	11.0	10.9	9.9	10.1	13.3	11.2	13.8	11.0
DE, Mcal/kg	3.14	3.10	3.14	3.04	3.85	3.96	3.72	2.08
NE_L_, Mcal/kg	1.54	1.53	1.54	1.50	1.54	2.09	1.97	1.65

^†^ DM: Dry matter of fresh matter, CP: Crude protein, RDP: Rumen Degradable Protein, NDF: Neutral detergent fiber, ADF: Acid detergent fiber, IVDMD: In vitro dry matter digestibility, Ca: Calcium, P: Phosphorus, ME: Metabolizable energy, DE: Digestible energy, NE_L_: Net energy required for lactation, G/RC: Grazing red clover with perennial ryegrass, G/WC: Grazing white clover with ryegrass annual, C/CR: Cut-and-carry red clover with perennial ryegrass, C/WC: Cut-and-carry white clover with ryegrass annual, CC: Commercial concentrate, GMG: Ground Maize grain, WDG: Wet Distiller Grains, and MS: Maize silage.

**Table 4 animals-11-00946-t004:** Estimation of supplement and forage intake in live metabolic weight (g/kg LW^0.75^ cow/day) using different strategies, five different supplementations (S) in dairy cattle under two feeding systems (grazing vs. cut-and-carry) with two leguminous forages (red clover or white clover).

Supplement Strategies	Supplement g/kg LW^0.75^	Forage, g/kg LW^0.75^	DMI, g/kg LW^0.75^
Supplementation (S)			
S1	0.0 ^d^	74.9	74.9 ^c^
S2	42.2 ^a^	74.9	117.1 ^a^
S3	31.1 ^c^	74.9	106.0 ^b^
S4	41.1 ^ab^	74.9	116.0 ^a^
S5	39.4 ^b^	74.9	114.3 ^a^
SEM	0.850		1.41
*p*-Value	<0.01		<0.01
Management (M)			
Cut-and-Carry	31.5 ^a^	74.9	106.4
Grazing	30.0 ^b^	74.9	105.0
SEM	0.538		0.890
*p*-Value	<0.01		0.112
Variety (Var)			
White Clover	30.8	74.9	105.7
Red Clover	30.8	74.9	105.7
SEM	0.316		0.338
*p*-Value	0.538		0.890
Overall	30.8	74.9	105.7
SEM	1.410		1.650
*p*-Value			
S × M	<0.01		0.342
S × Var	1.000		1.000
M × Var	0.646		0.774
S × M × Var	0.984		0.997

SEM: standard error of means. ^a, b, c, d^ values with a row with different superscripts differ significantly at *p* < 0.05.

**Table 5 animals-11-00946-t005:** Energy (MJ), crude protein, rumen degradable protein (g/cow day) of different strategies, five different simulated supplementations (S) in dairy cattle feed under two feeding systems (grazing or cut-and-carry), and two forage crops (red clover or white clover).

Supplement Strategies	ME (MJ)	CP (g/d)	RDP (g/d)	NDF (g/d)	Ca (g/d)	P (g/d)
Supplementation (S)						
S1	85.0 ^a^	1320.0 ^d^	792.0 ^c^	3655.1 ^d^	62.8 ^c^	29.4 ^c^
S2	135.3 ^b^	2175.0 ^a^	1305.0 ^a^	4883.6 ^a^	121.8 ^a^	46.1 ^a^
S3	122.8 ^c^	1960.6 ^b^	1176.4 ^b^	4575.7 ^b^	107.0 ^b^	41.9 ^b^
S4	144.7 ^a^	1900.0 ^bc^	1210.5 ^b^	4156.3 ^c^	66.9 ^c^	48.1 ^a^
S5	144.7 ^a^	1792.9 ^c^	1210.5 ^b^	4351.7 ^bc^	50.3 ^d^	45.5 ^a^
SEM	3.38	46.9	29.0	104.0	1.99	1.13
*p*-Value	<0.01	<0.01	<0.01	<0.01	<0.01	<0.01
Management (M)						
Cut and Carry	124.3 ^b^	1709.8 ^b^	1075.4 ^b^	4578.5 ^a^	83.7 ^a^	42.6
Grazing	128.7 ^a^	1949.6 ^a^	1202.3 ^a^	4070.5 ^b^	79.8 ^b^	41.8
SEM	2.14	29.6	18.9	65.7	1.26	0.714
*p*-Value	<0.01	<0.01	<0.01	<0.01	<0.01	0.307
Variety (Var)						
White Clover	126.6	1861.0 ^a^	1155.2	4268.8	82.3	42.0
Red Clover	126.4	1798.4 ^b^	1122.5	4380.2	81.2	42.4
SEM	2.14	29.6	18.9	65.7	1.26	0.714
*p*-Value	0.952	<0.01	0.085	0.091	0.052	0.536
Overall	126.5	1829.7	1138.9	4324.5	81.75	42.19
SEM	2.21	30.7	19.4	75.9	2.46	0.69
*p*-Value						
S × M	0.589	<0.01	<0.01	0.636	0.908	<0.01
S × Var	1.000	0.847	0.736	0.998	0.996	0.946
M × Var	0.749	0.373	0.475	<0.01	<0.01	<0.01
S × M × Var	0.999	0.995	0.987	0.991	0.958	0.968

EM: Energy metabolizable, CP: Crude protein, RDP: Rumen Degradable Protein, NDF: Neutral detergent fiber, Ca: Calcium, and P: Phosphorus. SEM: standard error of means. ^a, b, c, d, e^ Values with a row with different superscripts differ significantly at *p* < 0.05.

**Table 6 animals-11-00946-t006:** Balance (g/d) and metabolizable energy (MJ/d) of different strategies, five different simulated supplementations (S) in dairy cattle grazing under two feeding strategy systems management (grazing vs. cut-and-carry), and two legume varieties (red clover or white clover).

Supplement Strategies	ME (MJ/d)	CP (g/d)	RDP (g/d)	Ca (g/d)	P (g/d)
Supplementation (S)					
S1	0.0 ^c^	304.4 ^b^	81.1 ^a^	39.4 ^c^	10.9 ^c^
S2	−9.3 ^b^	445.7 ^a^	94.5 ^a^	82.0 ^a^	13.3 ^b^
S3	−21.9 ^a^	231.4 ^c^	−34.1 ^c^	67.2 ^b^	9.1 ^d^
S4	0.0 ^c^	170.7 ^d^	0.00 ^b^	27.0 ^d^	15.4 ^a^
S5	0.0 ^c^	63.6 ^e^	0.00 ^b^	10.4 ^e^	12.7 ^b^
SEM	1.09	11.3	8.30	1.05	0.34
*p*-Value	<0.01	<0.01	<0.01	<0.01	<0.01
Management (M)					
Cut-and-Carry	−7.7 ^b^	132.0 ^b^	-29.0 ^b^	47.4 ^a^	13.1 ^a^
Grazing	−4.8 ^a^	354.3 ^a^	85.6 ^a^	43.0 ^b^	11.5 ^b^
SEM	0.69	7.14	5.25	0.66	0.21
*p*-Value	<0.01	<0.01	<0.01	<0.01	<0.01
Variety (Var)					
White Clover	−6.2	274.2 ^a^	44.5 ^a^	45.7	12.2
Red Clover	−6.3	212.1 ^b^	12.1 ^b^	44.6	12.4
SEM	0.69	7.14	5.25	0.66	0.21
*p*-Value	0.901	<0.01	<0.01	0.093	0.516
Overall	−7.4	8.1	183.9	38.4	11.0
SEM	0.89	8.8	16.7	2.2	0.35
*p*-Value					
S × M	<0.01	<0.01	<0.01	0.414	<0.01
S × Var	1.000	<0.01	<0.01	0.962	<0.01
M × Var	0.510	<0.01	<0.01	<0.01	<0.01
S × M × Var	0.957	0.244	0.188	0.697	<0.01

SEM: standard error of means. ^a, b, c, d, e^ values with a row with different superscripts differ significantly at *p* < 0.05.

**Table 7 animals-11-00946-t007:** Emissions of methane (CH_4_)) of different strategies, five different supplementations (S) in dairy cattle under two feeding strategy management systems (grazing vs. cut-and-carry), and two legumes varieties (red clover or white clover).

Supplement Strategies	CH_4_ (g d^−1^)	CH_4_ (g kg^−1^ FCM 3.5%)
Supplementation (S)		
S1	253.55 ^c^	52.87 ^a^
S2	415.48 ^a^	25.74 ^b^
S3	375.04 ^b^	22.83 ^c^
S4	364.72 ^b^	22.23 ^c^
S5	422.58 ^a^	25.88 ^b^
SEM	9.29	0.84
*p*-Value	<0.01	<0.01
Management (M)		
Cut and Carry	368.5	31.0
Grazing	364.1	28.9
SEM	5.87	0.53
*p*-Value	0.452	<0.01
Variety (Var)		
White Clover	359.9 ^b^	30.4
Red Clover	372.7 ^a^	29.5
SEM	5.87	0.53
*p*-Value	<0.01	0.094
Overall	366.3	29.9
SEM	5.86	1.87
*p*-Value		
S × M	0.245	<0.01
S × Var	0.180	<0.01
M × Var	0.117	<0.01
S × M × Var	0.394	<0.01

FCM 3.5% (kg/d) = (0.432 × milk yield (kg)) + (16.23 × milk fat yield (kg)); SEM: standard error of means. ^a, b, c,^ values with a row with different superscripts differ significantly at *p* < 0.05.

**Table 8 animals-11-00946-t008:** Correlations among nutrient intake, emissions of methane (CH_4_) of different feeding strategies in dairy cattle in small dairy farmers under different feeding management systems.

Item	LW^0.75^ (kg/cow)	DMI (g/kg)	NDFI (g/d)	PC Intake (g/d)	ME Intake (g/d)	MY (kg/d)	FCM (kg/d)	CH_4_ (g d^−1^)	CH_4_ (g kg^−1^ FCM 3.5%)
LW^0.75^ (kg/cow)		0.596	0.682	0.543	0.543	0.425	0.433	0.527	−0.136
DMI (g/kg)	0.001		0.821	0.869	0.978	0.938	0.947	0.931	−0.740
NDFI (g/d)	0.001	0.001		0.704	0.698	0.746	0.749	0.828	−0.479
CP Intake (g/d)	0.001	0.001	0.001		0.833	0.849	0.857	0.810	−0.691
ME Intake (g/d)	0.001	0.001	0.001	0.001		0.919	0.929	0.901	−0.745
Milk Yield (kg/d)	0.001	0.001	0.001	0.001	0.001		0.989	0.896	−0.832
FCM (kg/cow/d)	0.001	0.001	0.001	0.001	0.001	0.001		0.904	−0.853
CH_4_ (g d^−1^)	0.001	0.001	0.001	0.001	0.001	0.001	0.001		−0.676
CH_4_ (g kg^−1^ FCM 3.5%)	0.019	0.001	0.001	0.001	0.001	0.001	0.001	0.001	

LW: live weight, DMI: Dry matter intake, NDFI: Neutral detergent fiber Intake, CP: Crude protein, and fat corrected milk, FCM 3.5% (kg/d) = (0.432 × milk yield (kg)) + (16.23 × milk fat yield (kg)). The upper part shows the absolute values, and the lower part shows the degree of significance (*p*-value).

**Table 9 animals-11-00946-t009:** Correlations among dry matter intake (DMI, kg/d) using different feeding strategies, five different supplementations (S) in dairy cattle under two feeding management systems (grazing or cut-and-carry), and two forage crops (red clover or white clover) compared with emissions of methane (CH_4_) and carbon dioxide (CO_2_).

DMI (kg/d)	CH_4_ (g d^−1^)	CH_4_ (g kg^−1^ FCM 3.5%)
S1	−0.990 ***	−0.370 ***
S2	0.991 ***	−0.916 ***
S3	0.997 ***	−0.798 ***
S4	0.992 ***	−0.668 ***
S5	0.784 ***	−0.648 ***
Cut−and−Carry	0.972 ***	−0.736 ***
Grazing	0.892 ***	−0.779 ***
Whit Clover	0.966 ***	−0.736 ***
Red Clover	0.905 ***	−0.770 ***

DMI: Dry matter intake. Fat corrected milk, and FCM 3.5% (kg/d) = (0.432 × milk yield (kg)) + (16.23 × milk fat yield (kg)); *** *p* < 0.001.

## Data Availability

The datasets generated during and/or analysed during the current study are available from the corresponding author on reasonable request.

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
