# Peer review of "Feeding Forage Mixtures of Ryegrass (*Lolium* spp.) with Clover (*Trifolium* spp.) Supplemented with Local Feed Diets to Reduce Enteric Methane Emission Efficiency in Small-Scale Dairy Systems: A Simulated Study"

_animals, 2021, doi:10.3390/ani11040946_

Round 1

Reviewer 1 Report

Dear Authors,

I have read your submission with high interest and even the issue isn't totally new becomes everyday of higher interest not only for scientific community but moreover for the public. 

In present, when dairy industry is under public pressure due to high proportion on GHG production, such research is important to provide breeders/ farmers with scientifically based advices on towards sustainable carbon/ nitrogen neutral management. Feeding management and diet composition based on local components and sources is one way to decrease carbon/ nitrogen footprint of farm. 

I have to thank you for quality of presentation in methodology as well in results. It was accurate, brief and clear, with referring important literature sources in the particular area. 

I have detect only small typo or grammar errors but as non-native speaker I would leave it for the proofreading stage. 

reviewer

Author Response

Reviewer 1: Comments and Suggestions for Authors

I have read your submission with high interest and even the issue isn't totally new becomes everyday of higher interest not only for scientific community but moreover for the public. 

In present, when dairy industry is under public pressure due to high proportion on GHG production, such research is important to provide breeders/ farmers with scientifically based advices on towards sustainable carbon/ nitrogen neutral management. Feeding management and diet composition based on local components and sources is one way to decrease carbon/ nitrogen footprint of farm. 

I have to thank you for quality of presentation in methodology as well in results. It was accurate, brief and clear, with referring important literature sources in the particular area. 

I have detect only small typo or grammar errors but as non-native speaker I would leave it for the proofreading stage. 

R: Thank you., for your   comments, the manuscript has been   revised by a native English speaker and    comments has been attended  in reference to the  other 3 reviewers

Reviewer 2 Report

Introduction is short and clearly structured, however, it is better if the authors more clearly highlight why their work is novel in comparison to previous publications. Moreover, it is necessary to introduce more general overview of the topic, including previous questions, objectives, results of similar research in the area,why authors focus on different types of management and legumes, and why this research is important to the scientific and society.

Materials and Methods require an appropriate experimental description, such as the chemical composition of commercial concentrate. The number of animals is not enough to support the data in the manuscript.

Author Response

Reviewer 2: Comments and Suggestions for Authors

Introduction is short and clearly structured, however, it is better if the authors more clearly highlight why their work is novel in comparison to previous publications. Moreover, it is necessary to introduce more general overview of the topic, including previous questions, objectives, results of similar research in the area,why authors focus on different types of management and legumes, and why this research is important to the scientific and society.

Materials and Methods require an appropriate experimental description, such as the chemical composition of commercial concentrate. The number of animals is not enough to support the data in the manuscript.

R: The study is important because there are few records of methane emissions in small-scale milk production systems. In fact, in the area of work there is only one record of methane emissions (Pozo-Leyva et al. 2019), which talks about nitrogen efficiency in the study region.

It is therefore recommended that animals graze pastures associated with legumes in order to reduce feed costs and cover their nutritional requirements, however, they often do not cover their energy or protein requirements, and hence farmers look for supplementation strategies with commercial concentrates or local by-products, which reduce costs and allow them to cover their nutritional requirements throughout the year.

The chemical composition of the commercial concentrate is described in table 2 and   explanation  and  how  was determined has been include it .

One of the characteristics of small-scale milk production systems is the small number of animals available to producers, as described in lines 90-94, which is why the number of animals used in the experiments is low, however, in the study region, work has been carried out with few experimental units and they are published in indexed journals, for example Plata-Perez et al. 2020, published in the Indian Journal of Animal Sciences; Vega-García et al. 2020, published in the Chilean Journal of Agicultural Research; Gomez-Miranda et al. 2020, published in the Canadian Journal of Animal Science. In this case we have used 15 experimental units in case we consider using a larger number of replicates in the future, thank you for your comments.

Reviewer 3 Report

Dear authors.

This submitted manuscript entitled "Feeding Forage Mixtures of Ryegrass (Lolium spp.)  with Clover (Trifolium spp.) Supplemented with Local Feed Diets to Reduce Enteric Methane Emission Efficiency in Small-Scale Dairy Systems: An Environmentally Friendly Alternative" has some merit to readers.

However, this manuscript has lack of information including animal IACUC information, forages sampling and handling, chemical analysis methods, no replicates for grazing animals and much more.

Author Response

This submitted manuscript entitled "Feeding Forage Mixtures of Ryegrass (Lolium spp.)  with Clover (Trifolium spp.) Supplemented with Local Feed Diets to Reduce Enteric Methane Emission Efficiency in Small-Scale Dairy Systems: An Environmentally Friendly Alternative" has some merit to readers.

However, this manuscript has lack of information including animal IACUC information, forages sampling and handling, chemical analysis methods, no replicates for grazing animals and much more.

R: The present simulation with dairy cows, and research was performed previously with collaborating farmers [8,9, 15, 16] followed the procedures of the Universidad Autónoma del Estado de México.To determine the diet specifications, the following variables were kept constant: LW, days of lactation, milk yield and chemical composition (Table 1) in all treatments. The data were obtained from previous research studies in the region [8,9, 15, 16]. Briefly, samples of pastures (red clover (RC) (Trifolium pratense) with perennial ryegrass (Lolium perenne) and white clover (WC) (Trifolium repens) with annual ryegrass (Lolium multiflorum), commercial concentrate, ground maize grains (GMG) and wet distillers’ grains (WDG), were dried in a forced-air oven at 60 °C for 48 h. Once dried, they were ground with a Wiley mill (2.0 mm screen; Arthur H. Thomas, Philadelphia, PA, USA), and analyzed in duplicates for DM (930.15), and nitrogen (N; 990.02) using the Association of Official Analytical Chemists (AOAC) [17] standard methods. Neutral detergent fiber (NDF) and acid detergent fiber (ADF) were determined following Van Soest et al. [18] methods. In vitro dry matter digestibility (IVDMD) was performed of each ingredient by the Ankom-Daiy method [19,20]. Other details on diet preparation, including the control and supplementation strategies, were described previously [8,9, 15, 16]

Reviewer 4 Report

Line 76: Are these annual average temperature and precipitation values?

Lines 114-117: When I read this at first I though you were saying that the G and C swards were different e.g. the G sward was RC and the C sward was WC. The next sentence was then confusing. Can you re-word to make this clearer?

Author Response

Q: Line 76: Are these annual average temperature and precipitation values?

R: Figure 1 is attached, showing the precipitation values and temperature averages for the last years (2011-2019).

The simulation study was based from experience in the municipality of Aculco, State of México, México (between 20° 05’ 57” N and 99º 49’ 41” W, 2428 m above sea level) with a sub humid temperature climate. The mean temperature during the years 2011-2019 was 16 °C, with a mean minimal temperature of 7°C and mean maximal temperature of 24°C, and 888 mm of rainfall (Figure 1).

Figure 1. Rainfall and mean temperatures during the years 2011-2019

Tmax: Maximal Temperature (ºC), Tmin: Minimal temperature (ºC) and rainfall ( mm)  in the  study area. The data were obtained from the meteorological station "La Concepcion" (No. 15189) run by CONAGUA-DGe.

Q: Lines 114-117: When I read this at first I though you were saying that the G and C swards were different e.g. the G sward was RC and the C sward was WC. The next sentence was then confusing. Can you re-word to make this clearer?

R: Revised as requested.

The diets were formulated using two pasture managements (M): grazing of cultivated pastures (G), which has been a successful technology adopted by farmers to reduce feeding costs [15] and cut-and-carry (C), a conventional feeding strategy used by the SSDS [16]. The two pastures managements (G and C) were composed of different clover varieties (Var): RC (Trifolium pratense) with perennial ryegrass (Lolium perenne) and WC (Trifolium repens) with annual ryegrass (Lolium multiflorum). The four combined diets were then formulated with different supplementation strategies and evaluated using the followed combinations: G/RC, G/WC, C/RC, and C/WC, using 15 cows per diet

Round 2

Reviewer 2 Report

I appreciated the revisions made to the paper, and note that authors have conducted a well-designed field study with valuable implications for animal production management.  Therefore, I believe
this study could be published. 

Author Response

Thank you very much for  your suggestions,    the manuscript  has been    improved  including  Table  1  with the all  previous studies that  we consider for    model the present  study  ( as a recommendation for  reviewer  3), thanks in advance

Reviewer 3 Report

1) I checked all the previous references in this study used as a model refences as he reported in the text (8, 9, 15, 16), but I haven’t found clear experimental design make five scenarios of supplements. The most of previous studies were investigated with dominant perennial ryegrass pasture with unknown white clover and some other mixed pastures (a mixture of perennial ryegrass (Lolium perenne cv. Bargala), annual ryegrass (Lolium multiflorum cv. Maximus), cocksfoot or orchard grass (Dactylis glomerata cv. Potomac) and white clover (Trifolium repens cv. Ladino). However, this study clearly designed with red clover vs. white clover pasture comparisions. In addition, I haven’t found any supplementation of wet distillers’ grains, but this study used it. 

2) Authors used cut and carry vs. grazing animals experimental design, however, breeds of dairy cows were totally mixed breed with very limited number of animals. One study used Holstein, but other studies were used various mixed breeds (Jersey, three were Jersey × Holstein and three were Brown Swiss × Holstein) with very limited number of animals used ranges from 2-3 cows. 

Conclusions: In the present study, datasets used by Celis-Alvarez et al. cannot be used/published because unknown sources of data. May be artificially data sets they used it. If authors would like to resubmit this manuscript, authors have to described in details and make Tables including diets, cattle breeds, forage species, number of animals, references etc.

Author Response

Reviewer 3

Thank you for the concerns! These concerns make us think deeper and accordingly revise the manuscript.

The novelty of the study is, we aimed to evaluate through simulation the effects of different dairy cow diets grazing two different pastures management and two varieties of legumes (red clover vs. white clover) with the inclusion as supplements of common local ingredients used in SSDS on the emission intensity of CH4 per kilogram of milk and on nitrogen and phosphorus intake and excretion.

We have revised the manuscript and hope the novelty of the study is apparent to readers.

1) I checked all the previous references in this study used as model references as he reported in the text (8, 9, 15, 16), but I haven’t found a clear experimental design makes five scenarios of supplements. The most of previous studies were investigated with dominant perennial ryegrass pasture with unknown white clover and some other mixed pastures (a mixture of perennial ryegrass (Lolium perenne cv. Bargala), annual ryegrass (Lolium multiflorum cv. Maximus), cocksfoot or orchard grass (Dactylis glomerata cv. Potomac) and white clover (Trifolium repens cv. Ladino). However, this study clearly designed with red clover vs. white clover pasture comparisons. In addition, I haven’t found any supplementation of wet distillers’ grains, but this study used it. 

R Table 1 has been included for a better understanding of why these scenarios and others such as supplementation with wet distillers' grains have been suggested, due to the costs and time involved in conducting an in vivo study with the number of replicates (animals) and the homogeneity of the study, the best options were considered as supplementation strategies to understand which scenario may be best under our conditions.

2) Authors used to cut and carry vs. grazing animals experimental design, however, breeds of dairy cows were totally mixed breed with a very limited number of animals. One study used Holstein, but other studies were used various mixed breeds (Jersey, three were Jersey × Holstein and three were Brown Swiss × Holstein) with a very limited number of animals used ranges from 2-3 cows. 

The criterion of standardizing the number of cows, the breed, and their milk production, was so that there would be no significant differences between animals and milk production and the results would focus on feeding strategies, based on previous studies (Tables 1, 2). The number of animals was increased to increase the number of replicates, from previous studies (Table 1), and at the suggestion of the reviewers, the number of replicates was increased to 15 animals per treatment.

Conclusions: In the present study, datasets used by Celis-Alvarez et al. cannot be used/published because of unknown sources of data. Maybe artificially data sets they used it. If authors would like to resubmit this manuscript, authors have to describe in detail and make Tables including diets, cattle breeds, forage species, number of animals, references etc.

R Table 1  has been included, according to your suggestion, and  giving the support   and the reason for performing the present study

Round 3

Reviewer 3 Report

This manuscript is not clear described M & M, especially experimental design.

Thanks,

Author Response

Dear reviewer, thank you for your suggestions, we have added in the title the note that it is a simulation study, also we have included in the discussion a little more about the importance of the study and similar results. 

P16  L 308 

Marín-Santana et al. [26] in their study, founded similar milk yields (19.16 kg milk yields), than the present study, grazing ryegrass pastures and kikuyu grass (Pennisetum clandestinum) associated with white clover. In another study [22], carried out in dairy cows feed with Ryegrass grazing Festulolium associated with white clover, oat silage, and commercial concentrate obtained similar milk yields (18.86 kg/d/cow ) than in the present study .It is important to consider that under the conditions of the present simulation study, the predictions made with previous studies coincide [17, 21,22,23,24,26], considering that under SSDS systems, they are in the range of 14 to 20 kg milk production.

P15 L364  

Regardless of management cut and carry or grazing (366±2g/day) was similar between treatments (P>0.05), being lower when using white clover (359.9 g/day) compared to red clover (372.7 g/day), the latter agrees with Carrillo Hernandez et al. [23] who found that cows that grazed annual pastures emitted a higher amount of methane (266.6 g/day), compared to cows that grazed perennial pastures (242.6 g/day), hence, it is important to consider that depending on the type of forage, enteric fermentation and therefore methane emission will vary.

This manuscript is a resubmission of an earlier submission. The following is a list of the peer review reports and author responses from that submission.

Round 1

Reviewer 1 Report

no comments

Author Response

I would like to thank you for your comments, and accept   the present version, also   we  attend the  comments  of the reviewes  2 and 3,  which has been  adresed.

Please note that changes   of the reviewers  2 and  3 are colored in red

Sincerely  the authors

Reviewer 2 Report

Dear Author,
I am glad to revisit this interesting research article. The Submission has been substantially improved and is worthy of publication. As mentioned in the review comments, SSDS has limitations in designing of experiment but in case of scientific publication I would expect that some minimum requirements for use of normal statistics would be respected, i.e. more animals per. Scenario x management x composition. 6 cows are not enough even in the case of using non-parametric tests of mean. This is the main limitation of the study.

Author Response

Reviewer 2: Comments and Suggestions for Authors

Dear Author,

I am glad to revisit this interesting research article. The Submission has been substantially improved and is worthy of publication. As mentioned in the review comments, SSDS has limitations in designing of experiment but in case of scientific publication I would expect that some minimum requirements for use of normal statistics would be respected, i.e. more animals per. Scenario x management x composition. 6 cows are not enough even in the case of using non-parametric tests of mean. This is the main limitation of the study

R: Thank you very much for your suggestion, the consideration of the 6 cows was based on herd size, the structural characteristics of the small farmers dairy systems in Mexico herds. herd size range among 3 up to 20 animals (Espinoza-Ortega et al., 2005). Martínez-García et al. (2015) in their study consider this criteria (Data were collected from 115 farmers, the sample size represents 5% of the total farms in the study area), founding an average herd size of 10 cows, of which only 4.4 cows were in dairy production, for this  reason  we consider  6 dairy cows per   treatment, in future studies  we will consider  more animals as  you suggest in order to minimize  our  animal variation.

There are also other countries where the characteristics of Small dairy farmers system production herds have a similar herd size than  our study, in the central region of Colombia, the herds reported an average herd size around 20 animals, considering in all their reproductive stages (Nivia et al. 2018) ( dry, lactating and calves) o in  East-Central Africa, report  a  herd between 1 to 3 cows in dairy production in Smallholder Intensive system or up to  10 dairy cows in small-holder extensive system (Ndambi et al 2007).

Sincerely 

The authors

Reviewer 3 Report

In the introduction section, the authors should mention the relationship between the structure and composition of diet and production of methane. Furthermore, the study lacks a hypothesis. Please, state the hypothesis of the study based on the background and indicate in the conclusion if your hypothesis was proved or not. The authors should mention the relationship between the structure and composition of diet and production of methane. In addition, please add the innovation point, significance, and purpose of your study in the introduction section. Especially in the MM section, the authors should directly describe the experimental design, sample collection and detection, and statistical analysis in detail.

Author Response

REPONSE TO REVIEWERS

Reviewer 3: Comments and Suggestions for Authors

  1. In the introduction section, the authors should mention the relationship between the structure and composition of diet and production of methane. Furthermore, the study lacks a hypothesis. Please, state the hypothesis of the study based on the background and indicate in the conclusion if your hypothesis was proved or not. The authors should mention the relationship between the structure and composition of diet and production of methane. In addition, please add the innovation point, significance, and purpose of your study in the introduction section. Especially in the MM section, the authors should directly describe the experimental design, sample collection and detection, and statistical analysis in detail.

R Thank you for   your suggestions, we include the hypothesis   of the study, and were included in the conclusion.

In the  introduction Given this context, the hypothesis addressed in the present study was that it is feasible to reduce the emission of CH4 and CO2 eq emissions per kilogram of milk and reduce the excretion of nitrogen and phosphorus, and thus reducing the environmental footprint of these systems.

In the conclusion, The hypothesis is accepted as supplements from local inputs can fulfill the nutritional requirements of dairy cattle at moderate production levels along with the use of grazing pastures

As you suggest. The authors should mention the relationship between the structure and composition of diet and production of methane, In this sense we perform a correlation between     DM, NDF, CP and  ME intake, Milk yield , FCM 3.5%,   with   CH4  (g/d-1) and kg CO2 eq/d-1. Lines 251-272, 375-380, 441-447, (Table 7 ), and also   according to the  literature there  are a correlation  between DM intake  and CH4  emission,  in this sense  we perform correlations among dry matter intake (DMI, kg/d) using different feeding strategies, five different supplementations (S) in dairy cattle under two feeding systems (grazing or cut-and-carry), and two forage crops (red clover or white clover) compared with emissions of methane (CH4) and carbon dioxide (CO2)  (Table 8.)   and   were included in the discussion.

Discussion section

Similar to our study, Renand et al. [35] found in beef Heifers a positive correlation between DMI with CH4 and CO2 eq.. Methane is a byproduct of anaerobic microbial fermentation of feed in the rumen, and energy used for its synthesis is considered as a loss of energy for animal production; it has been calculated that the energy loss fluctuates between 3% and 6.5% on average for cattle fed diets high in concentrates and low‐quality pastures, respectively [20]. Inclusion of supplementation strategies in the diets resulted in a significant increase (p < 0.01) in CH4 production. Methane increase were 88%, 56%, 59 % and 80%, compared S1 vs S2, S3, S4 and S5, respectively (where 100% is equivalent to the emission of CH4 measured without the inclusion of supplements, S1 as a control diet), and coincides with Bell et al. [36] the average CH4 concentration calculated from burping peaks was positively associated with TDMI.

Lassen et al. [37] in a study with Holstein dairy cows found a positive genetic correlation (P<0.05) between milk yield and CH4 production, which suggests the relationship between energy intake, CH4 production and milk production, which agrees with the present results, Hegarty et al. [38] reported a 40% lower daily DMI and an analogous 25% lower methane production (g/d) between high and low-residual feed intake in steers, which is the difference in daily DMI (15%) obtained between the high- and low intakes in heifers.

In the present study, S1 present 40 % lower DMI (g/kg LW0.75) compared with the rest of supplementation strategies, which corresponds with these lower CH4 production. Total daily CH4 production and kg CO2 eq/d-1 is higher in S2 to S5 than S1, this is normal because the forage proportion in the basal diets is higher than S1 (forage: concentrate ratio was 100:0, S1). It is well established that both daily CH4 emission and kg CO2 eq/d- per unit of DMI increase as a result of increased NDF intake [39]. The present results coincide with Bittante and Cecchinato [40], who correlated milk yield, which is genetically determined, with estimated daily methane production, although this is mainly because the latter is obtained from the former and because increased milk yield imply larger DMI and rumen loads.

Sincerely 

The authors
